# Spatial Pattern Simulation of Antenna Base Station Positions Using Point Process Techniques

**Stelios Zimeras**

Department of Statistics and Actuarial-Financial Mathematics, University of the Aegean, 83200 Samos, Greece; zimste@aegean.gr

**Abstract:** Spatial statistics is a powerful tool for analyzing data that are illustrated as points or positions in a regular or non-regular state space. Techniques that are proposed to investigate the spatial association between neighboring positions are based on the point process analysis. One of the main goals is to simulate real data positions (such as antenna base stations) using the type of point process that most closely matches the data. Spatial patterns could be detailed describing the observed positions and appropriate models were proposed to simulate these patterns. A common model to simulate spatial patterns is the Poisson point process. In this work analyses of the Poisson point process—as well as modified types such as inhibition point process and determinantal Poisson point process—are presented with simulated data close to the true data (i.e., antenna base station positions). Investigation of the spatial variation of the data led us to the spatial association between positions by applying Ripley's K-functions and L-Function.

**Keywords:** point process; Poisson point process; antenna base stations; inhibition point process; determinantal Poisson point process; Ripley's K-function; L-Function

## 1. Introduction

Spatial statistics refers to the study of spatially combined data using appropriate statistical methodologies. In spatial statistics, point pattern process is a powerful tool to analyze the spatial association between different positions of a particular event (locations of the antenna base stations) and is considered an advanced statistical technique based on the homogeneity of the point process space. Homogeneity of the point process is explained by the dynamic effects under the spatial association leading us to simulate the appropriate spatial patterns [1–3].

The main goal of point pattern analysis is to investigate the process and especially to examine if the pattern is independent, regular or clustered. Based on these patterns, simulations is considered, and analysis of the final patterns takes place, so as to investigate their closeness to the real data [4,5]. For that reason, appropriate spatial models must be introduced to simulate the real data; the most common models are the Poisson point process and modifications such as inhibition point process and determinantal Poisson point process. The first model considers the distance between two positions using a proposed thresholding variable δ which is used to delete the positions close to δ. The second model considers the diversity between two positions, in the sense that the probability of observing two points close to or similar to each other is lower than the Poisson point process [6].

Many times in point process analysis, the prediction of the positions of the appropriate event is considered, and appropriate simulations are applied to overcome inaccuracies. In our case, this approach is applied, where our goal is to propose point process models close to the real data and simulate them for prediction of the positions. A variety of methods for the visualization of point data are provided based on the spatial patterns. These methods allow us to view the point patterns, explore structure in data by estimations from appropriate models, and test hypotheses relating to the process by considering the observed event distribution.

Finally the investigation of the spatial association for the proposed point process patterns could be considered, with the motive of explaining the regularity or in-regularity of the process, especially the existence, or degree of intensity, of the clustering patterns. In this work, investigation of spatial locations simulation is proposed based on Ripley's K-function and L-function.

## 2. Materials and Methods

*Point Process Analysis*

Considering a 2D space S where $S \subset R^d$ with d the dimension of the data, a region of interest could be introduced. This region could be representing rectangular or non-rectangular areas of investigation based on the particular analysis (Figure 1).

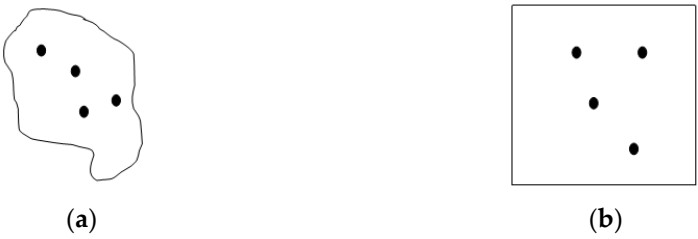

**(a)**　　　　　　　　　　　　　　　**(b)**

**Figure 1.** (**a**) Non-rectangular and (**b**) Rectangular regions of interest.

Data could be presented inside the 2D region in a coordinate format denoted as (x,y). Every coordinate consists of a corresponding position s; z(s) is denoted as the value for each z in position s. The z(s) is defined as a random variable at each position and the spatial model corresponding to the random variable is denoted as $\{\mathbf{z}(\mathbf{s}) : \mathbf{s} \in S\}$. The data consists of n positions $s_1 \ldots s_n$, each based on an (x,y) coordinate in 2D space. Pattern analysis, based on the spatial associations between positions, is represented by the following types: 1. independent, 2. regular and 3. clustered (Figure 2) [7–9].

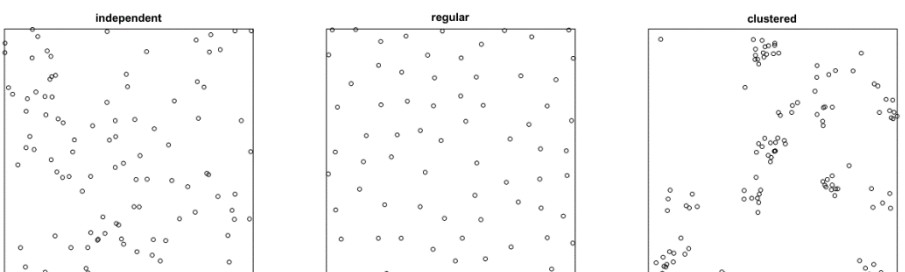

**Figure 2.** Types of spatial patterns.

The **Poisson random process** with distribution is given by Equation (1) [10]:

$$p[N(A)] = \frac{\lambda a(A)}{k!} e^{-\lambda a(A)}, k = 0, 1, 2, \ldots \tag{1}$$

where $\lambda$ is the value of the intensity and $a(.)$ is the unit area for a specific region and $A \in S \subset R^2$. The random variable $N(A)$ is the number of events in the set $A \subset X$ defined as a random process (Figure 3a).

A point process **Z(s)** is called a **determinantal** random point process if for a random subset $A$ drawn according to Z(s), the process holds for every $A \subset X$ (Equation (2)):

$$p[A \subset X] = \det(K) \tag{2}$$

where $K = \left[K_{ij}\right]_{ij \in A}$ is the matrix with rows and columns indexed by the elements of the set $A$. In our case, the point process is based on the form $p[N(A) \subset X] = \det(K)$, where $p[N(a)]$ is given by the Poisson random point process. Alternatively, a Poisson point process

where distance between two positions is taking place could be considered by defining it as an **inhibition process** [10] with distance δ, where δ is the threshold distance by deleting all pairs of events that are closer than δ (Figure 3b). Finally a point process with diversion called **determinantal point process [11–13]** is illustrated in Figure 3c. A determinantal point process is often used to induce diversity or repulsiveness among the points of a sample. Figure 3a illustrates simulated coordinates (x,y) of 100 positions of a **Poisson point process** with estimated $\widehat{\lambda} = 2.018$ simulated on the unit square. Figure 3b illustrates simulated coordinates (x,y) of 100 points of an **inhibition point process** with estimated $\widehat{\lambda} = 0.781$ and δ = 0.1 simulated on the unit square. Figure 3c illustrates simulated coordinates (x,y) of 100 points of a **determinantal point process** with estimated $\widehat{\lambda} = 3.01$ simulated on the unit square.

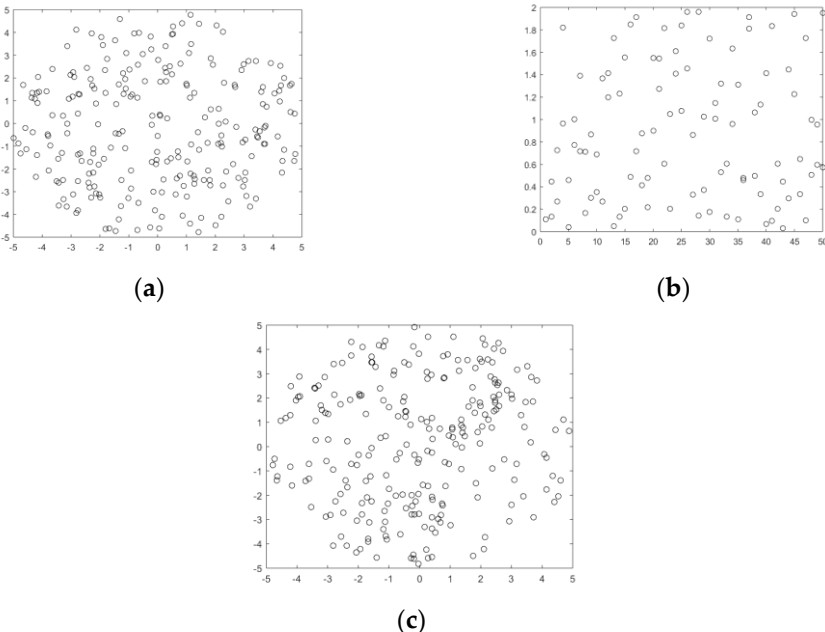

(a)

(b)

(c)

**Figure 3.** Point process types: (**a**) Poisson; (**b**) inhibition Poisson; and (**c**) determinantal.

The main properties for the Poisson process are [11–13]:

- The number of positions in a region A has a Poisson distribution with mean $\lambda N(A)$
- The positions of these points are i.i.d. and uniformly distributed inside *A*
- The contents of two disjoint regions A and B are independent

Consider the locations of base stations, i.e., antennas, of the mobile network in Paris [14] considering all the base stations, of all operators and for all operating frequencies (Figure 4). Based on the previous point process types, it is clear that the real data are close to the Poisson point process leading to a similar simulating pattern.

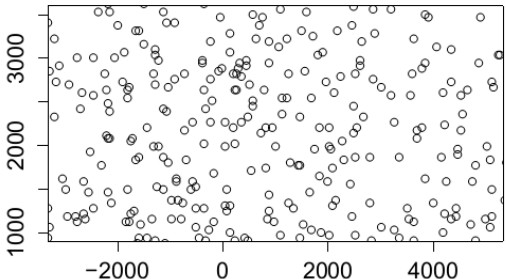

**Figure 4.** Real data locations of base stations, antennas of the mobile network in Paris [14].

Based on the Poisson point process, the mean number of events per unit area at the point *s*, defined as $\lambda(s)$, is given by [15–18] Equation (3):

$$\lambda(s) = \lim_{ds \to 0} \left\{ \frac{E(N(ds))}{|ds|} \right\} \tag{3}$$

where $d(s)$ is a small region around the point *s*, $E(.)$ is the expected value, and $N(d(s))$ is the number of events in the small region, defined as the first-order property considering a study region. Then, the expected value over the region *A* is given by Equation (4):

$$E[N(A)] = \lambda |A| \tag{4}$$

where *A* is the area of the sub-region, and $\lambda$ is the value of the intensity.

When the analysis involves two study regions, in this case a second order property would be considered given by [15–18] Equation (5):

$$\gamma(s_i, s_j) = \lim_{ds_i, ds_j \to 0} \left\{ \frac{E(N(ds_i)N(ds_j))}{|ds_i||ds_j|} \right\} \tag{5}$$

where, under the condition of independence between the two regions A and B, the reflection of the spatial dependence in the process taking place is given by Equation (6):

$$\text{cov}[N(A), N(B)] = 0 \text{ if } A \cap B = \emptyset) \tag{6}$$

A stationary process is defined when the intensity inside the region A is constant, so $\lambda(s) = \lambda$ and $\gamma(s_i, s_j) = \gamma(s_i - s_j) = \gamma(d)$ (depending only on direction and distance). For the isotropy case, the second-order intensity depends only on the distance between $s_i$ and $s_j$. A brief presentation of the previous research involving point process analysis in antenna stations is given in Table 1 [19–23].

**Table 1.** Presentation of previous research into random point process involving antenna stations for telecommunications.

| References | Brief Presentation |
|---|---|
| Aurélien Vasseur (2017) [14] | Focus on Poisson point process considering probabilistic modeling using data based on antennas of the mobile network in Paris |
| Yingzhe Li, François Baccelli, Harpreet S. Dhillon, Jeffrey G. Andrews (2014) [19] | Poisson random point analysis using determinant point process |
| Ezequiel Fattori, Pablo Groisman, and Carlos Sarraute (2016) [20] | Modeling the spatial distribution of cell phone antennas in the city of Buenos Aires (CABA) |
| Benedikt Jahnel (2018) [21] | Probabilistic methods in Telecommunications |
| A. Guo, Y. Zhong, M. Haenggi, and W. Zhang (2014) [22] | Modeling using Gauss–Poisson point process |
| N. Deng, W. Zhou, and M. Haenggi (2015) [23] | Modeling using Ginibre point process |

## 3. Results

Analysis of the spatial point pattern is to consider the neighboring structure of the region of interest based on distances between the positions. An appropriate measure to calibrate the spatial dependence is Ripley's K-function given by the form [24–27] (Equation (7)):

$K(d) = \lambda^{-1} E$ [number of extra points within distance d of a randomly chosen arbitrary point] (7)

where the expected number of event positions under the distance d is $\lambda \pi d^2$, defining $\lambda$ as value of the intensity under Poisson random point process; so $K(d) = \pi d^2$. The value $\lambda K(d)$ could be defined as the expected number of points within distance d of a randomly chosen arbitrary point; hence (Equation (8)):

$$\lambda \widehat{K}(d) = \frac{1}{n} \sum_{i=1}^{n} (\text{number of events within distane d of the event i}) \tag{8}$$

The number of event positions could be estimated by an indicator function $I_d(d_{ij})$. Thus, estimation of the K-function is given by the form (Equation (9)):

$$\widehat{K}(d) = \frac{1}{\lambda n} \sum_{i=1}^{n} \sum_{j=1}^{n} I_d(d_{ij}) \tag{9}$$

The unbiased estimate of the K-function can be finally written by [19–22] Equation (10):

$$\widehat{K}(d) = \frac{1}{\lambda n} \sum_{i=1}^{n} \sum_{\substack{j=1 \\ j \neq i}}^{n} w_{ij} I_d(d_{ij}) \tag{10}$$

where: $d_{ij}$ is the distance between the $i$-th and $j$-th observed event position, which can be viewed as the radius of a circle centered at event position $i$ and passing through $j$; and $w_{ij}$ is a weighting value equal to this circle's proportion of the entire area A. For specific distance value $d$, $I_d(d_{ij})$ is an indicator function which is 1 if $d_{ij} \leq d$. It is clear that as $d$ increases, $w_{ij} \to \infty$. If $K(d) > \pi d^2$, the theoretical K-function, then there is evidence of clustering. Investigation for a clustering pattern could be achieved by plotting $\widehat{K}(d)$ and $K(d)$, so spatial dependences could be illustrated. Figure 5b graphically compares Ripley's K-function and the theoretical one, and clearly shows evidence of regularity, as well as the Poisson random point process with 1000 positions (Figure 5a) [28–30].

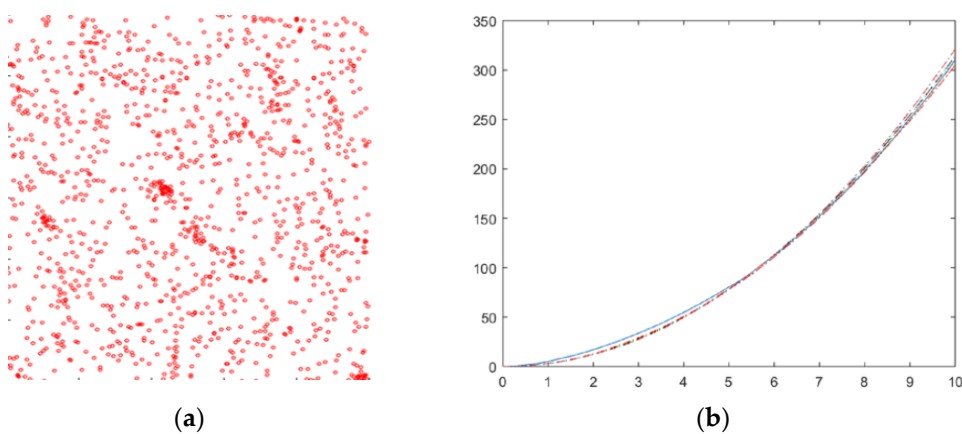

(a) (b)

**Figure 5.** (**a**) Poisson random point process; and (**b**) Ripley's K-function.

This plot is given in Figure 5b, where we see possible evidence for clustering, because the observed K-function is above the curve corresponding to a random process (theoretical one); where blue is the observed K-function, black dashed is the theoretical K-function, and red dashed are the lower and upper limits for K-Function. Especially at small scales ($d < 5$), the process does not show evidence of clustering. At other scales ($d > 5$), we have evidence of clustering. The lower and upper limits for the K-functions are given by $L(d) = \min\left\{\widehat{K}(d)\right\}$ and $U(d) = \max\left\{\widehat{K}(d)\right\}$.

Another approach, based on the K-function, is to transform using $L(d) = \sqrt{K(d)/\pi}$ or $L(d) = \sqrt{\frac{K(d)}{\pi}} = d$, where peaks of positive values in a plot would correspond to clustering for the corresponding scale $d$ [19–22]. Figure 6b represents the graph between the L-function and the theoretical one, and clearly shows evidence of regularity, as well as the Poisson random point process with 1000 positions (Figure 6a) [28–30].

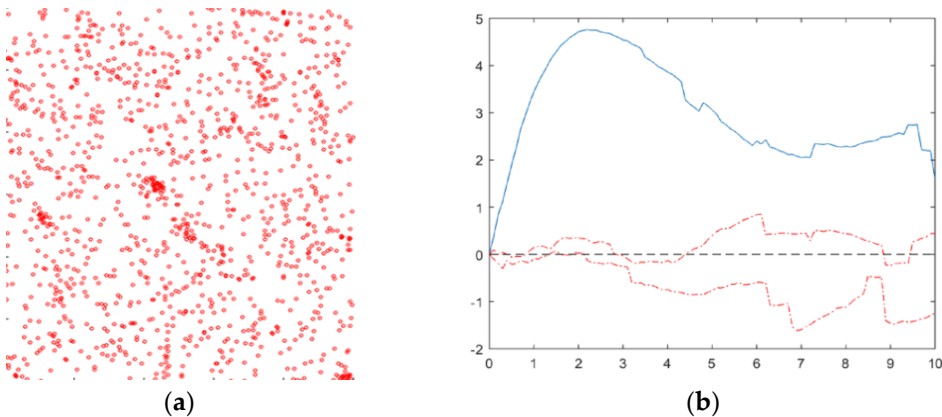

(a)           (b)

**Figure 6.** (**a**) Poisson random point process; and (**b**) L-function.

In the L-function graph, the clustering is better illustrated, with a blue line (observed L-function) above the black dashed line corresponding the theoretical one; red dashed lines are the lower and upper limits for L-Function with $L(d) = \min\left\{\widehat{L}(d)\right\}$ and $U(d) = \max\left\{\widehat{L}(d)\right\}$. Since the L-function lies above the upper limit, the clustering is significant.

## 4. Conclusions

Point process analysis is widely used for the investigation of the spatial dependences or associations between different positions in a region of interest. The proposed techniques are especially useful in cases when spatial patterns need to be simulated, and modeled realizations of the spatial process represent the real data as best they can. One of these cases is the situation when the reproduction of the antenna positions are needed to be investigated, and spatial patterns are produced by simulations of the specific type of point process, such as the Poisson point process.

In this work, analysis of the Poisson point process is presented, and pattern realizations of the proposed process are illustrated by simulation data, explaining as closely as they can the real antenna positions. Investigation of the spatial association between position distances considering neighborhood structure is undertaken by using Ripley's K-function and L-function. Application of these distance functions proves useful in the explanation of the simulated data and by extension for the real data.

**Funding:** This research received no external funding.

**Institutional Review Board Statement:** Not applicable.

**Informed Consent Statement:** Not applicable.

**Data Availability Statement:** Not applicable.

**Conflicts of Interest:** The author declare no conflict of interest.

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
