# Peer review of "Spatial Pattern Simulation of Antenna Base Station Positions Using Point Process Techniques"

_telecom, doi:10.3390/telecom3030030_

Round 1
Reviewer 1 Report
This paper presents "Spatial pattern simulation of Antenna base station positions using point process techniques" but there are some comments that need to be addressed before the acceptance
- The author should further ellaborate the novelty of the proposed work
- It would be more great if the author will incorporte a table highlighting the comparison with the previous research.
- Author did not numbered the equations. Please cite the equations in the text as well.
-Author need to improve the quality of the figures
- Author cited very old references, please use the latest references
Author Response
Dear Sir
Appologies for the delay due to health problem. In red are the changes
Thanks in advance
Assoc. Prof. S. Zimeras

Reviewer 2 Report
The paper needs some major improvements before publication:
1. The manuscript needs to be carefully checked because there are many grammatical errors and typos.
2. There are quantities which are not properly defined such as equation 73, equation 114, equation 118 and equation 137.
3. The contributions are not clear enough. For example, it is not obvious whether the data used in the results section are real data or just random locations.
4. It is not obvious whether the model will fit all the areas or specific area such us a dense urban area or suburban area or rural area.
5. Analysis and simulations is little, the author did not discuss the antenna deployments which is very important in this scenario. We do not know whether it is a traditional grid model or random BS model or actual 4G network.
Author Response
Dear Sir
Appologies for the delay due to health problems. In red are the changes
Many thanks
Assoc. Prof. S. Zimeras

Reviewer 3 Report
Reviews
The idea of “Spatial pattern simulation of Antenna base station positions using point process techniques” is good to be introduced in antennas as a base station that investigates the spatial variation of the data leading to the spatial association between the positions by applying Ripley K-functions and L-Function. But the following comments should be acknowledged.
· Reference [15] is missing in section 2.
· There are many grammatical mistakes, for example, in lines 36, 42, 83, etc.
· In section 2, Materials and methods (subsection 2.1 Point process analysis), is not defined in Poisson Point Process (line 73). but later in the same section, represents the value of intensity (line 119). In section 3, (Results) is used in another equation as a Poisson parameter. Please, justify how’s possible in the same that the one symbol like representing many parameters like the Poisson parameter and value of intensity.
· The First formula in section 3, will confuse to readers, as it is not written properly.
· In the results section, the figures are not properly labeled. Also, improve the quality of figures.
· References are not properly cited in the text, as the second paragraph from section 2 is from ref [18] but cited with [7-9]. The reference of the determinantal point process is cited in the text as [10], but it is in the references section as [11].
The overall plagiarism from Turnitin is 36%, and the plagiarism from the first source is 8%, and not cited.
Author Response
Dear Sir
Appologies for the delay due to health problem. In red are the changes
Many thanks
Assoc. Prof. S. Zimeras
